# Exploring the molecular structures that confer ligand selectivity for galanin type II and III receptors

Yoo-Na Lee[1], Arfaxad Reyes-Alcaraz[1,2], Seongsik Yun[1], Cheol Soon Lee[3], Jong-Ik Hwang[1], Jae Young Seong [1]*

1 The GPCR laboratory, Graduate School of Biomedical Science, Korea University College of Medicine, Seoul, Republic of Korea, 2 College of Pharmacy, University of Houston, Houston, Texas, United States of America, 3 Graduate School of Biomedical Science, Korea University College of Medicine, Seoul, Republic of Korea

* jyseong@korea.ac.kr

**Data Availability Statement:** All relevant data are within the paper and its Supporting Information files.

## Abstract

Galanin receptors (GALRs) belong to the superfamily of G-protein coupled receptors. The three GALR subtypes (GALR1, GALR2, and GALR3) are activated by their endogenous ligands: spexin (SPX) and galanin (GAL). The synthetic SPX-based GALR2-specific agonist, SG2A, plays a dual role in the regulation of appetite and depression-like behaviors. Little is known, however, about the molecular interaction between GALR2 and SG2A. Using site-directed mutagenesis and domain swapping between GALR2 and GALR3, we identified residues in GALR2 that promote interaction with SG2A and residues in GALR3 that inhibit interaction with SG2A. In particular, Phe$^{103}$, Phe$^{106}$, and His$^{110}$ in the transmembrane helix 3 (TM3) domain; Val$^{193}$, Phe$^{194}$, and Ser$^{195}$ in the TM5 domain; and Leu$^{273}$ in the extracellular loop 3 (ECL3) domain of GALR2 provide favorable interactions with the Asn$^5$, Ala$^7$, Phe$^{11}$, and Pro$^{13}$ residues of SG2A. Our results explain how SG2A achieves selective interaction with GALR2 and inhibits interaction with GALR3. The results described here can be used broadly for *in silico* virtual screening of small molecules for the development of GALR subtype-specific agonists and/or antagonists.

## Introduction

G-protein-coupled receptors (GPCRs) are a superfamily of membrane proteins with more than 860 members in humans [1]. GPCRs are responsible for a variety of physiological functions including growth, homeostasis, reproduction, sleep, appetite, mood behavior, and others. Because of their diverse roles, GPCRs represent the largest family of therapeutic targets in human medicine [2, 3]. GPCRs are modulated by various endogenous ligands including peptides, amino acids, lipids, and nucleotides [4–8]. The characterization of crystal structures of agonist/antagonist-bound GPCRs provides crucial clues for the development of synthetic agonists and antagonists [9, 10]. In addition, site-directed mutagenesis [11–13] has yielded

**Funding:** This work was supported by the Research Program (NRF-2015M3A9E7029172) of the National Research Foundation (https://ernd.nrf.re.kr/index.do) funded by the Ministry of Science, ICT and Future Planning, Korea. The funders had no role in study design, data collection and analysis, decision to publish or preparation of the manuscript.

**Competing interests:** The authors have declared that no competing interests exist.

insights into ligand-receptor interactions and has helped identify binding sites between ligands and GPCRs.

The galanin receptors (GALR1, GALR2, and GALR3) and their corresponding peptide ligands galanin (GAL) and spexin (SPX) emerged simultaneously through a local gene duplication followed by whole-genome duplications [8, 14]. Because of the overall high degrees of amino acid sequence similarity between the galanin receptors, the two distinguishable ligands, GAL and SPX, exhibit cross-reactivity to the three receptor subtypes, albeit with different mechanisms of action [14, 15]. For instance, SPX activates GALR2 and GALR3, whereas GAL exhibits relatively high affinity for GALR1 and GALR2 but not GALR3 [14]. Both SPX and GAL are able to bind to GALR2 and induce G-protein-mediated signaling. However, after the signal transduction, SPX only marginally induces GALR2 internalization, whereas GAL induces substantial GALR2 internalization, leading to initiation of an alternative signaling pathway [16, 17]. Thus, unlike GAL, SPX is an endogenous biased agonist that preferentially activates G-protein-mediated signaling but not internalization-mediated signaling [15, 18]. Those differences in receptor preference and mechanisms of action might account for the opposing effects that SPX and GAL have on appetite and reproductive behaviors [19–22].

In addition to the differences in ligand cross-reactivity, the GALRs activate different G-protein signaling pathways. GALR1 and GALR3 induce $G_i$-coupled inhibitory signaling, whereas GALR2 induces $G_q$-coupled stimulatory signaling [23]. Thus, it is possible that SPX could simultaneously induce stimulatory G-protein signaling via GALR2 and inhibitory G-protein signaling via GALR3. The same simultaneous activation of opposing G-protein signaling pathways might occur with GAL-mediated GALR1 and GALR2 activation. The complexity of those ligand/receptor-mediated signal transduction systems highlights the need for a better understanding of the pathophysiological functions of each GALR subtype, which might lead to the development of GALR subtype-selective agonists and antagonists as therapeutic agents for GALR-associated diseases [23–25].

In an earlier study, we developed SPX-based GALR2-specifc agonists (SG2A) [26]. Like SPX, SG2A interacts with GALR2 to induce $G_q$-coupled stimulatory signaling with little internalization of the receptor, but it lacks activity toward GALR3 [15]. Treatment of cortisol-induced depression-like mice with SG2A led to a decrease in anxiety and depressive behaviors, likely via the activation of serotonergic neurons located in the dorsal raphe nucleus [27]. That result is in good agreement with previous reports of GALR2-mediated anxiolytic and anti-depressant effects [28, 29], suggesting that SG2A and SPX activate GALR2 similarly under pathophysiological conditions.

The full GALR2-specific activity of SG2A can be achieved through quadruple substitution of the SPX sequence with the $Asn^5$, $Ala^7$, $Leu^{11}$/ $Phe^{11}$, and $Pro^{13}$ residues derived from the corresponding positions in GAL, while mutations in other residues of SPX do not influence the GALR2/GALR3 specificity [26]. The SPX peptide with quadruple substitution is called the Qu-SPX peptide [26]. Those substitutions completely abolish the activity of the peptide toward GALR3, indicating that GAL-derived residues are critical for the ligand to distinguish between GALR2 and GALR3. Little is known, however, about which GALR2 residues interact with the GAL-derived residues of Qu-SPX, or which GALR3 residues are required for the Qu-SPX ligand to inhibit interaction with that receptor. Therefore, we sought to identify the GALR2 residues responsible for the specific interactions with Qu-SPX by using chimeric and point-mutated GALR2 and GALR3 proteins, with the long-term goal of providing information that will support the design of therapeutic agonists and antagonists that are specific for GALR2.

## Materials and methods

### Peptide synthesis

Human wild-type (WT) SPX (NWTPQAMLYLKGAQ-NH$_2$) and mutant SPX peptides with GALR2-specific residues (Qu-SPX, [N$^5$]-SPX, [A$^7$]-SPX, [F$^{11}$]-SPX, and [P$^{13}$]-SPX; Fig 1) were synthesized by AnyGen (Gwangju, Korea). The peptides were dissolved in distilled water as 10 mM stock solutions and stored at -80˚C until use.

### Plasmid DNA constructs

The pcDNA3.1 vector was purchased from Invitrogen (San Diego, CA, USA). The serum response element (SRE)-luciferase (SRE-luc) vector containing a single copy of the SRE (CCA-TATTAGG) conjugated with luciferase was purchased from Stratagene (La Jolla, CA, USA). The cDNAs for human GALR1, GALR2, and GALR3 were obtained from BRN SCIENCE, Inc. The cDNAs were inserted into the *EcoR*l and *Xho*l sites of pcDNA3.1.

### Construction of chimeric receptors and site-directed mutagenesis

For domain swapping between GALR2 and GALR3, individual cDNA fragments of interest were amplified by polymerase chain reaction (PCR) with *Pfu* polymerase (ELPIS Biotech, Daejeon, Korea) and two specific primers, one corresponding to the 5' or 3' end of the GALR2 or GALR3 cDNA and the other to the region of overlap between the two receptors. The two resulting fragments, one from GALR2 and the other from GALR3, were subjected to a second round of PCR to generate the chimeric cDNAs. To construct single, double-, triple-, pentuple-, hextuple-, and septuple-mutant receptors, amino acids of GALR3 were substituted for the amino acids at the corresponding positions in GALR2 using PCR-based site-directed mutagenesis. All of the chimeric constructs were cloned into the pcDNA3 expression vector at *Hind*lll and *Xba*l sites. The DNA sequences of the chimeras were verified by automatic sequencing.

### Cell culture and transfection

HEK293 cells stably expressing the G$_{qi}$ construct, which allows induction of G$_q$-dependent signaling pathways upon activation of a G$_i$-coupled receptor [14, 30], were maintained in

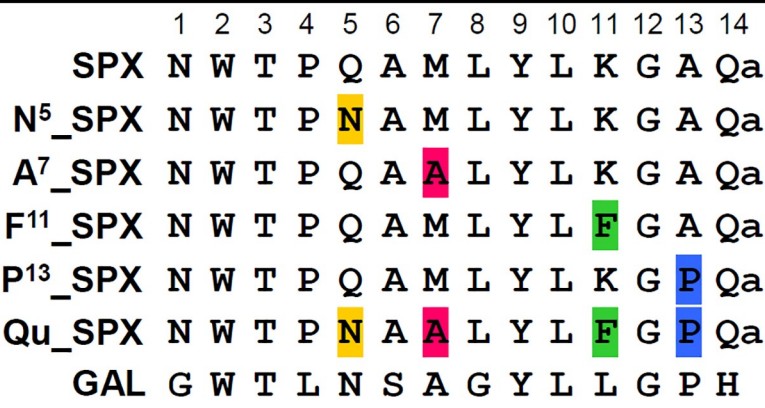

**Fig 1. Amino acid sequences of the SPX, mutant SPX, and GAL peptides.** Single and quadruple mutant peptides derived from SPX. The substituted amino acids (SPX to GAL) are indicated in different colors (N$^5$, Yellow; A$^7$, Pink; F$^{11}$, Green; and P$^{13}$, blue). For GAL, amino acid sequence at positions 15 to 30 (AVGNHRSFSDKNGLTS) after H at position 14 is not shown.

Dulbecco's Modified Eagle's Medium (DMEM) in the presence of 10% fetal bovine serum and 1% penicillin and streptomycin. For all transfections, 200 ng DNA (100 ng receptor and 100 ng SRE-luc) mixed with lipofectamine 2000 was applied to 60–80% confluent cells on a multi-well plate according to the manufacturer's instructions (Invitrogen, Carlsbad, CA).

### Luciferase assay

For luciferase assays, HEK293 $G_{qi}$ cells were seeded on 48-well plates at a density of $2.5 \times 10^4$ cells/well one day before transfection. A mixture including 100 ng SRE-luc reporter construct, 100 ng expression plasmid, and lipofectamine 2000 reagent in diluted Opti-MEM (Gibco) was incubated at room temperature for 20 min and then added into each well according to the manufacturer's instructions (Invitrogen, CA, USA). Before exposure to the ligands, the cells were maintained in serum-free DMEM for 16–18 h. Approximately 48 h after transfection, the cells were treated with ligand for 6 h. The cells were then lysed by the addition of 100 μl lysis buffer. The luciferase activity in 50 μl cell extract was determined using a luciferase assay system according to the standard protocol for the Synergy 2 Multi-Mode Microplate Reader (Bio-Tek, Winooski, VT, USA).

### Cyclic adenosine monophosphate accumulation

SPX-induced or Qu-SPX-induced cyclic adenosine monophosphate (cAMP) mobilization was measured in HEK293 cells stably expressing the pGlosensor[TM]-22F cAMP plasmid (Promega Corp., Madison, WI, USA). The Glosensor-22F cAMP HEK293 cells were seeded on 96-well plates at a density of $2.0 \times 10^4$ cells/well 24 h before transfection. A mixture including 100 ng expression plasmid and lipofectamine 2000 reagent in diluted Opti-MEM (Gibco) was incubated at room temperature for 20 min. The mixtures were added into each well according to the manufacturer's instructions (Invitrogen, CA, USA). After 48 h, Glosensor cAMP substrates were added to the transfected cells in $CO_2$-independent media. After 2 h, the cells were incubated with a range of agonist concentrations or vehicle for 10 min and then exposed to foskolin (10 μM). Luminescence was measured for up to 30 min using a Microplate Reader (BioTek, Winooski, VT, USA) [7].

### Data analysis

Data analysis was performed by non-linear regression with a sigmoidal dose-response curve. The concentrations of agonists that induced half-maximal stimulation ($EC_{50}$) were calculated using the GraphPad PRISM5 software (GraphPad software, Inc., San Diego, CA). All data are presented as the mean ± standard error (SE) of at least three independent experiments.

## Results

### Determination of the receptor domains that interact with Qu-SPX

To determine the Qu-SPX-interacting domains of the GALR2 receptor, a series of chimeric receptors were generated by domain swapping between GALR2 and GALR3. The GALR2/3 chimeric receptors had the N-terminal domain of GALR2 and the C-terminal domain of GALR3, whereas the GALR3/2 chimeric receptors had the N-terminal domain of GALR3 and the C-terminal domain of GALR2 (**Fig 2**). The membrane expression the chimeric receptors was measured using the Nano-Glo HiBit extracellular detection system [15], which showed that all the chimeric receptors were substantially expressed on the plasma membrane, albeit at different levels (**S1 Fig**). The GALR2/3 chimeric receptors did not respond to the Qu-SPX peptide; however, except for GALR2/3f, they did respond to SPX. The strong response to SPX was

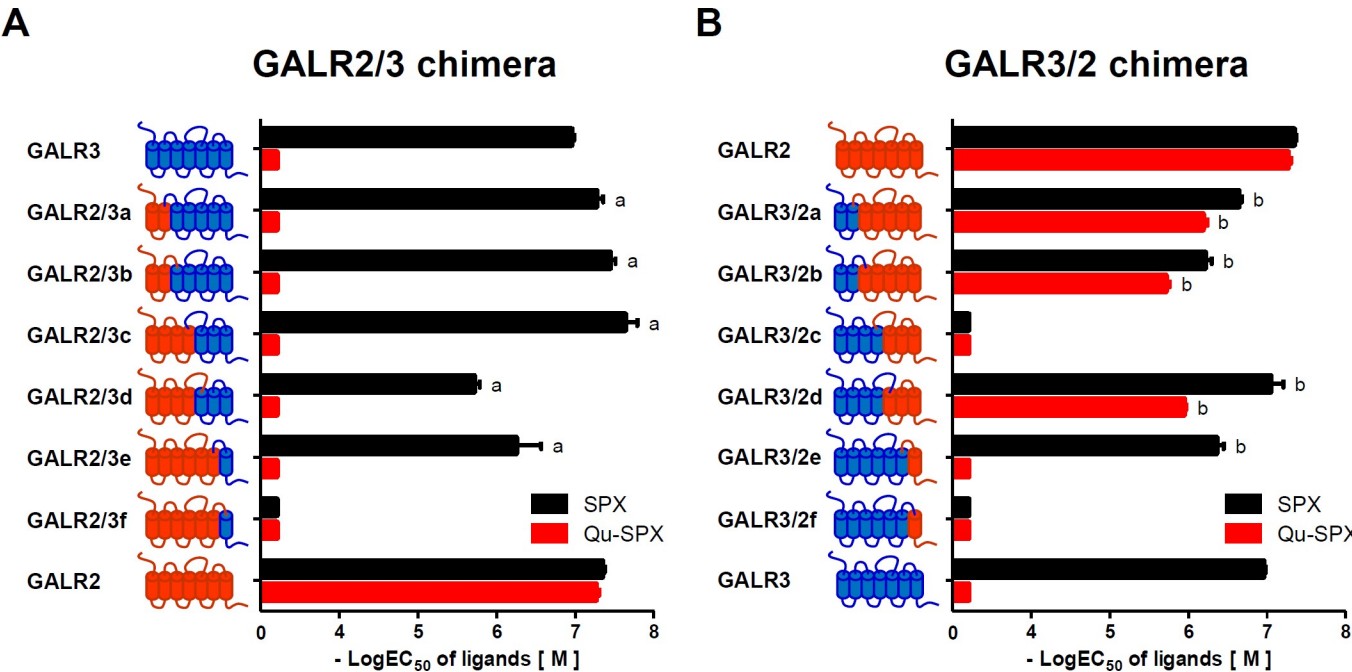

**Fig 2. Differential responses of chimeric receptors to SPX and the GALR2-specific agonist Qu-SPX.** A series of GALR2/3 and GALR3/2 chimeric receptors were constructed. (A) GALR2 domains were serially replaced with the corresponding domains of GALR3. (B) GALR3 domains were serially replaced with the corresponding domains of GALR2. The potencies of the responses of the chimeric receptors to SPX and Qu-SPX were examined in HEK293 cells stably expressing $G_{qi}$ protein. Forty-eight hours after transfection, the cells were treated with SPX or Qu-SPX for 6 h. Luciferase activities were then determined. The $EC_{50}$ values are presented as the mean ± SE (a: $P < 0.05$ vs. WT GALR3; b: $P < 0.05$ vs. WT GALR2).

retained by all of the GALR2/3 series a to c chimeric receptors, whereas it was drastically weakened in the GALR2/3 series d and e chimeric receptors (**Fig 2A** and **S1 Table**), indicating that the receptors retained GALR3-like behaviors when the GALR3 region from the N-terminus to the transmembrane helix 4 (TM4) domain was replaced by the corresponding GALR2 region.

For the GALR3/2 chimeric receptors, the GALR3/2a and GALR3/2b receptors showed decreased responses to SPX and Qu-SPX compared with those of WT GALR2, likely because of the longer length of the N-terminal portion of GALR3 compared with that of GALR2. GALR3/2c and GALR3/2f did not respond to either ligand. GALR3/2d responded to both SPX and Qu-SPX, whereas GALR3/2e responded to SPX but not to Qu-SPX (**Fig 2B** and **S1 Table**).

The ligand-response experiments revealed that: 1) some chimeric constructs (GALR2/3a, GALR2/3b, and GALR2/3c) were selective for SPX but not for Qu-SPX; and 2) GALR3/2e had much stronger selectivity than GALR3/2d for Qu-SPX. On the basis of those findings, we designed further experiments to identify the residues in GALR3 that hamper interaction with the Asn[5], Ala[7], Phe[11], and Pro[13] residues of Qu-SPX. In addition, by substituting amino acids from GALR2 into the corresponding positions in GALR3, we could identify the GALR2 residues that permit interaction Qu-SPX.

## Interactions between Asn[5] and Ala[7] of Qu-SPX and the TM3 domain of GALR2

To determine which transmembrane (TM) domain of GALR2 was responsible for the specific interactions with the substituted amino acids in Qu-SPX, the chimeric receptors that showed selective responses to SPX but not to Qu-SPX (GALR2/3a, GALR2/3b, and GALR2/3c) were exposed to single-residue SPX mutants ([N[5]]-SPX, [A[7]]-SPX, [F[11]]-SPX, and [P[13]]-SPX; **Fig**

3). GALR2 responded similarly to SPX and each of the single-residue mutants, whereas GALR3 responded about 10 times more strongly to SPX than to any of the single-residue mutants, which is in good agreement with our previous results [26]. Like GALR3, GALR2/3a and GALR2/3b responded to each of the single-residue mutants about 10 times less potently than they responded to SPX. By contrast, GALR2/3c responded to [N^5]-SPX and [A^7]-SPX about as strongly as it responded to SPX. Moreover, the responses of GALR2/3c to [N^5]-SPX and [A^7]-SPX were significantly stronger than those of GALR2/3b (**Fig 3**). On the other hand, all of the chimeric receptors responded weakly to [F^11]-SPX and [P^13]-SPX (**Fig 3** and **S2 Table**). Because the main difference between GALR2/3b and GALR2/3c is the TM3 and TM4 domains, the results suggested that either TM3 or TM4 was responsible for the selective interactions of GALR2/3 with [N^5]-SPX and [A^7]-SPX.

To address that issue, we constructed GALR3 mutants containing either the TM3 domain (GALR3/2_[TM3]) or the TM4 domain (GALR3/2_[TM4]) of GALR2 (**Fig 4A**). The responses of GALR3/2_[TM4] to [N^5]-SPX and [A^7]-SPX were similar to those of GALR2 and weaker than those of GALR3. By contrast, the responses of GALR3/2_[TM3] to [N^5]-SPX and [A^7]-SPX were similar to those of GALR2 and stronger than those of GALR3 (**Fig 4A** and **S3 Table**). These results indicate that the differences in the amino acid sequences of the TM3 domain between GALR2 and GALR3 are responsible for the selective responses to [N^5]-SPX and [A^7]-SPX.

To further identify which amino acids within the TM3 domain of GALR2 are important for the interactions with Asn^5 and Ala^7 of Qu-SPX, we compared the amino acid sequences of the TM3 domains of GALR2 and GALR3. The amino acids within the TM3 domain are well conserved between GALR2 and GALR3, but Phe^103, Phe^106, and His^110 in human GALR2 are changed to Leu^100, Tyr^103, and Tyr^107 at the corresponding positions in human GALR3 (**Fig 4B**). We generated single, double, and triple mutants of GALR3 by replacing those three residues of GALR3 with the corresponding residues from GALR2. The single-residue mutations did not significantly increase the responses of the mutant GALR3 proteins to [N^5]-SPX or [A^7]-SPX. Furthermore, the Tyr^107His single-mutant GALR3 did not respond to any ligand (**Fig 4C** and **S3 Table**). By contrast, the double-mutant and triple-mutant GALR3 proteins responded more strongly than WT GALR3 to both [N^5]-SPX and [A^7]-SPX. In particular, the Leu^100Phe, Tyr^107His double-mutant had a markedly stronger response than WT GALR3 to [A^7]-SPX. The triple-mutant GALR3 responded more strongly than any of the other GALR3 variants to [N^5]-SPX. These results suggest that Ala^7 mainly interacts with the Phe^103 and His^110 residues of GALR2, whereas Asn^5 interacts with all three (Phe^103, Phe^106, and His^110) residues of GALR2 (**Fig 4D** and **S3 Table**). Alternatively, it is possible that the Phe^103, Phe^106, and His^110 residues within the TM3 domain of GALR2 affect the conformation of the receptor in a way that allows increased interaction with the Asn^5 and Ala^7 residues of Qu-SPX.

## Interaction between Pro^13 of Qu-SPX and the TM5 domain of GALR2

We next tried to determine which GALR3/2 domains were responsible for the selective interactions with Phe^11 and/or Pro^13 of Qu-SPX. GALR3/2e did not respond at all to Qu-SPX, whereas GALR3/2d, which contained TM5/6 of GALR2, did respond to Qu-SPX (**Fig 2B**), suggesting that TM5/6 of GALR2 might contribute to the ligand selectivity. To investigate that possibility, we constructed chimeric receptors in which the TM5 or TM6 domains of GALR3 were replaced with the corresponding domains of GALR2, resulting in the GALR3/2_[TM5] and GALR3/2_[TM6] mutants, respectively (**Fig 5A**). GALR3/2_[TM5] responded more strongly than WT GALR3 to [P^13]-SPX but not to [F^11]-SPX. The response of GALR3/2_[TM5] to [P^13]-SPX was similar to the response of that mutant receptor to WT GALR2. These results indicate that Pro^13 of Qu-SPX might interact with the TM5 domain of GALR2. GALR3/2_[TM6] did not

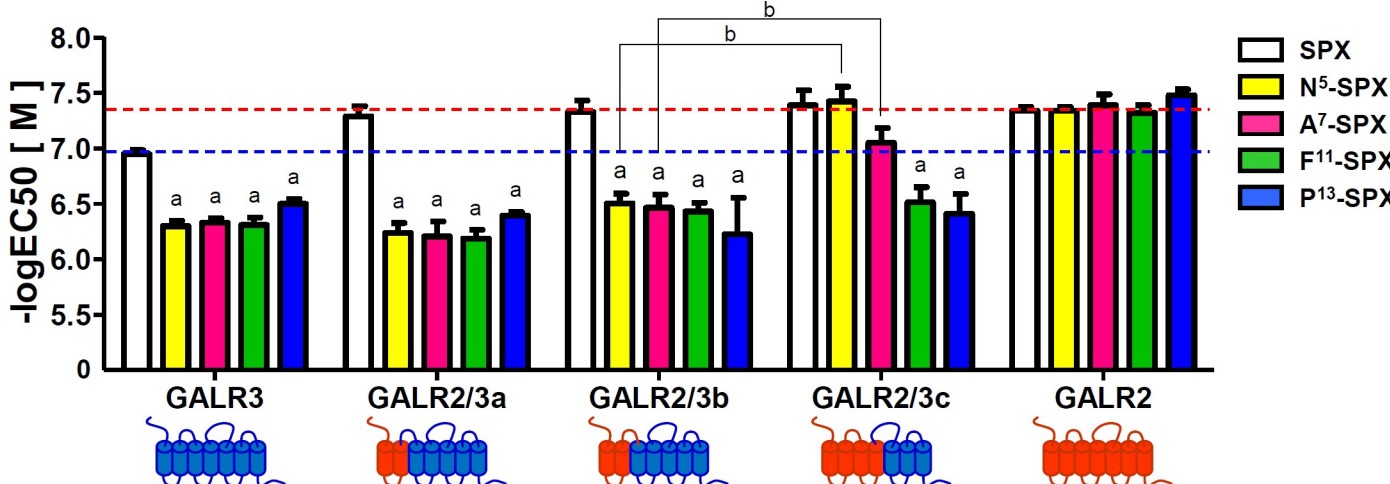

**Fig 3. Responses of the GALR3-like chimeric receptors to single-residue mutant ligands.** The potencies of single-substitution mutant peptides toward GALR3-like chimeric receptors were examined on the basis of luciferase activity. The horizontal red and blue dashed lines represent the potency of SPX toward GALR2 and GALR3, respectively. The $EC_{50}$ values are presented as the mean ± SE (a: $P < 0.05$ vs. SPX; b: $P < 0.05$ vs. the GALR2/3b chimeric receptor).

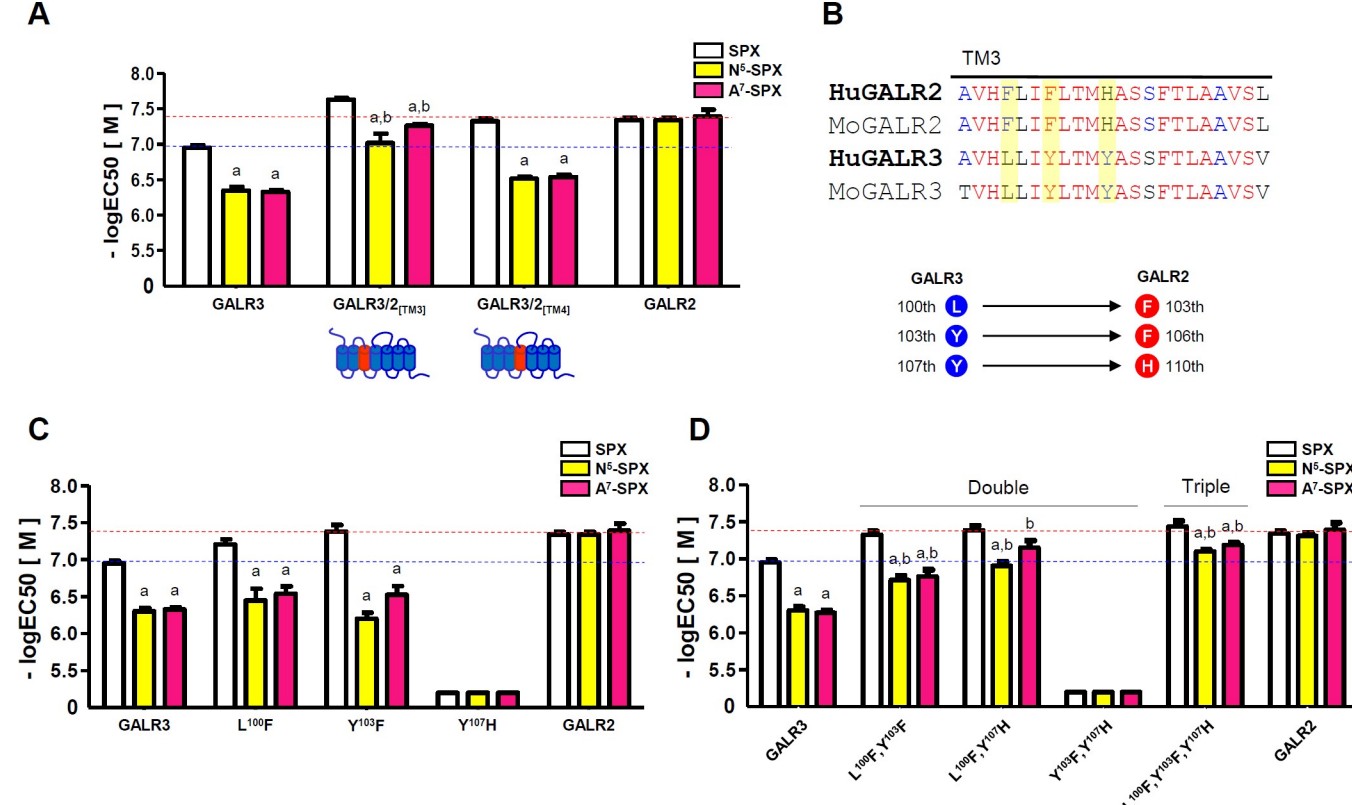

**Fig 4. Determination of the core regions in GALR2 that interact with the $Asn^5$ and $Ala^7$ residues of the ligand.** (A) The TM3 and TM4 domains of GALR3 were replaced with the corresponding domains of GALR2. The responses of the mutant receptors to SPX, [$N^5$]-SPX, and [$A^7$]-SPX were then tested. (B) Amino acid sequence comparison of the TM3 domains from human (Hu) and mouse (Mo) GALR2 and GALR3. (C) The responses of the single-mutant ($Leu^{100}Phe$, $Tyr^{103}Phe$, or $Tyr^{107}His$) GALR3 proteins to SPX, [$N^5$]-SPX, and [$A^7$]-SPX. (D) The responses of the double-mutant and triple-mutant GALR3 proteins to SPX, [$N^5$]-SPX, and [$A^7$]-SPX. The horizontal red and blue dashed lines represent the responses of GALR2 and GALR3, respectively, to SPX. The $EC_{50}$ values are presented as the mean ± SE (a: $P < 0.05$ vs. SPX; b: $P < 0.05$ vs. WT GALR3).

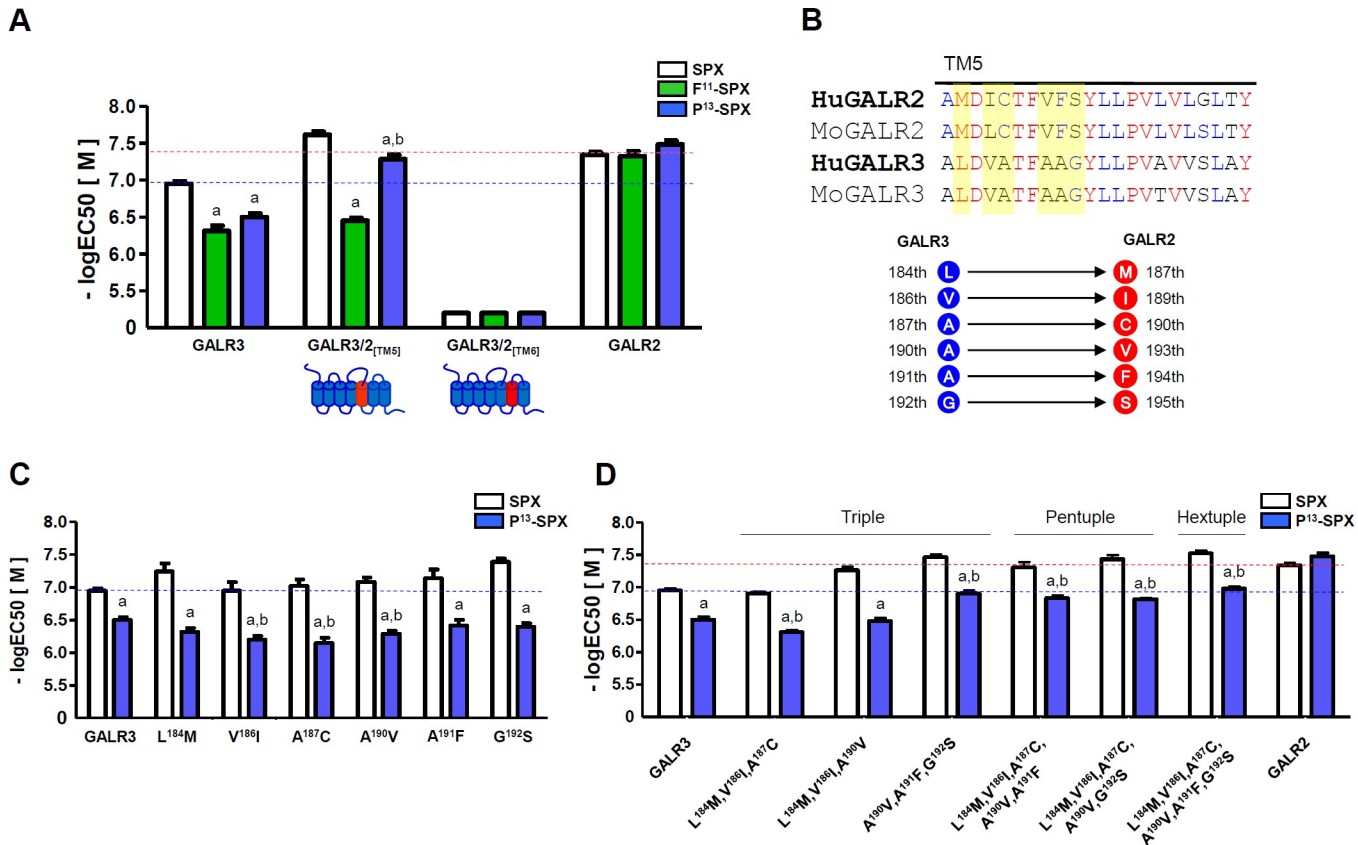

**Fig 5. Identification of amino acid residues within the TM5 domain that interact with Pro[13] of the ligand.** (A) Responses to SPX, [P[13]]-SPX, and [F[11]]-SPX by chimeric receptors in which the TM5 or TM6 domains of GALR3 were replaced with the corresponding domains of GALR2. (B) Amino acid sequence comparison of the TM5 domains between human (Hu) and mouse (Mo) GALR2 and GALR3. (C) Responses to [P[13]]-SPX by GALR3 variants with the Leu[184]Met, Val[186]Ile, Ala[187]Cys, Ala[190]Val, Ala[191]Phe, and Gly[192]Ser single substitutions in the TM5 domain. (D) Reponses to [P[13]]-SPX by GALR3 variants with triple, pentuple, and hextuple substitutions in the TM5 domain. The horizontal red and blue dashed lines represent the responses of GALR2 and GALR3, respectively, to SPX. The EC$_{50}$ values are presented as mean ± SE (a: $P < 0.05$ vs. SPX; b: $P < 0.05$ vs. WT GALR3).

respond to any ligand, suggesting that the TM6 domain might help to stabilize the conformational structure of the receptor (**Fig 5A** and **S4 Table**).

We compared the TM5 amino acid sequences between GALR2 and GALR3 to identify which amino acids might be important for the interaction with [P[13]]-SPX. The Met[187], Ile[189], Cys[190], Val[193], Phe[194], and Ser[195] residues of the TM5 domain of human GALR2 correspond to the Leu[184], Val[186], Ala[187], Ala[190], Ala[191], and Gly[192] residues of human GALR3, respectively. Other sequences within the TM5 domain are either well conserved or highly variable across vertebrate species (**Fig 5B**). We constructed GALR3 mutant receptors with single, triple, pentuple, and hextuple substitutions in which the GALR3 residues were replaced by the corresponding GALR2 residues (**Fig 5C and 5D**). All of the single-substitution receptors responded to [P[13]]-SPX with potencies similar to that of WT GALR3, suggesting that single amino acid substitutions are not enough to recover the full GALR2/ligand interaction (**Fig 5C** and **S4 Table**). The triple-mutant containing the Ala[190]Val, Ala[191]Phe, and Gly[192]Ser substitutions exhibited a significantly stronger response than WT GALR3 to [P[13]]-SPX, which was similar to the responses of the pentuple and hextuple mutants, suggesting that the Val[193], Phe[194], and Ser[195] residues of GALR2 are especially important for binding to the Pro[13] residue of Qu-SPX (**Fig 5D** and **S4 Table**).

## Interaction between the Phe[11] residue of the ligand and the ECL3 domain of GALR2

Finally, we investigated the GALR2 domains that might be responsible for the selective interaction with Phe[11] of Qu-SPX. Because we had already examined the TM domains in the receptors, we focused on the extracellular loop (ECL) domains of GALR2 and their potential interaction with Phe[11] of Qu-SPX. We replaced the ECL domains of GALR3 with those of GALR2, which resulted in the mutants GALR3/2[ECL1], GALR3/2[ECL2], and GALR3/2[ECL3] (**Fig 6A**). The GALR3/2[ECL3] mutant responded to [F[11]]-SPX as strongly as it did to SPX, suggesting that the ECL3 domain of GALR2 interacts with Phe[11] of Qu-SPX (**Fig 6A**).

To determine which amino acids within the ECL3 domain of GALR2 might contribute to an interaction with Phe[11] of Qu-SPX, we compared the ECL3 amino acid sequences between GALR2 and GALR3 (**Fig 6B**). We then constructed single mutants in which the Ala[264], Phe[265], Ser[266], Pro[267], and Cys[272] residues of the ECL3 domain of human GALR3 were replaced by the corresponding Pro[265], Leu[266], Thr[267], Arg[268], and Leu[273] residues, respectively, of human GALR2. We found that the response of the GALR3[Cys[272]Leu] mutant to [F[11]]-SPX was stronger than that of WT GALR3 and similar to that of WT GALR2. Those results indicated that the Leu[273] residue within the ECL3 domain of GALR2 plays an important role in the interaction with Phe[11] of Qu-SPX (**Fig 6C** and **S5 Table**).

## A GALR3 mutant receptor with GALR2-derived residues responds to Qu-SPX

Our results showed that seven amino acid residues within the TM3, TM5, and ECL3 domains of GALR2 are likely responsible for the interaction between that receptor and the Asn[5], Ala[7], Phe[11], and Pro[13] residues of Qu-SPX. Therefore, we constructed a septuple-mutant GALR3 harboring Leu[100]Phe, Tyr[103]Phe, and Tyr[107]His substitutions in the TM3 domain; Ala[190]Val, Ala[191]Phe, and Gly[192]Ser substitutions in the TM5 domain; and a Cys[272]Leu substitution in the ECL3 domain (**Fig 7A**). In the SRE-luc assay system [5], the septuple-mutant GALR3 responded to Qu-SPX in a dose-dependent manner, whereas WT GALR3 did not (**Fig 7B and 7C** and **S6 Table**). To corroborate those results, we performed a cAMP assay. GALR3 is coupled to the $G_i$ inhibitory signaling pathway. We therefore measured the inhibition of forskolin-induced cAMP production in cells expressing WT GALR3 or the septuple-mutant GALR3, in the presence of SPX or Qu-SPX. In WT GALR3-expressing cells, SPX inhibited the forskolin-induced cAMP levels in a dose-dependent manner, whereas Qu-SPX failed to do so (**Fig 7D** and **S6 Table**). In septuple-mutant GALR3-expressing cells, both SPX and Qu-SPX reduced the forskolin-induced cAMP levels (**Fig 7E** and **S6 Table**). Those results demonstrate that the Leu[100], Tyr[103], and Tyr[107] residues in the TM3 domain; the Ala[190], Ala[191], and Gly[192] residues in the TM5 domain; and the Cys[272] residue in the ECL3 domain of GALR3 are likely responsible for the lack of response to Qu-SPX, and that replacement of those residues with corresponding residues from GALR2 make the mutant receptor responsive to Qu-SPX. However, the partial recovery of Qu-SPX potency toward the septuple-mutant GALR3 needs to be further explained. Molecular dynamics-resolved structures of SPX and Qu-SPX differ each other, but both peptides may not have stable three-dimensional structures in solution (**S2 Fig**). Therefore, it is likely that conformations of Qu-SPX different from those of SPX may hamper full recovery of the potency toward the septuple-mutant GALR3.

## Discussion

Exploration of the amino acid residues in receptors that contribute to ligand binding and receptor activation provides a basis for optimal drug discovery. As a preliminary approach to

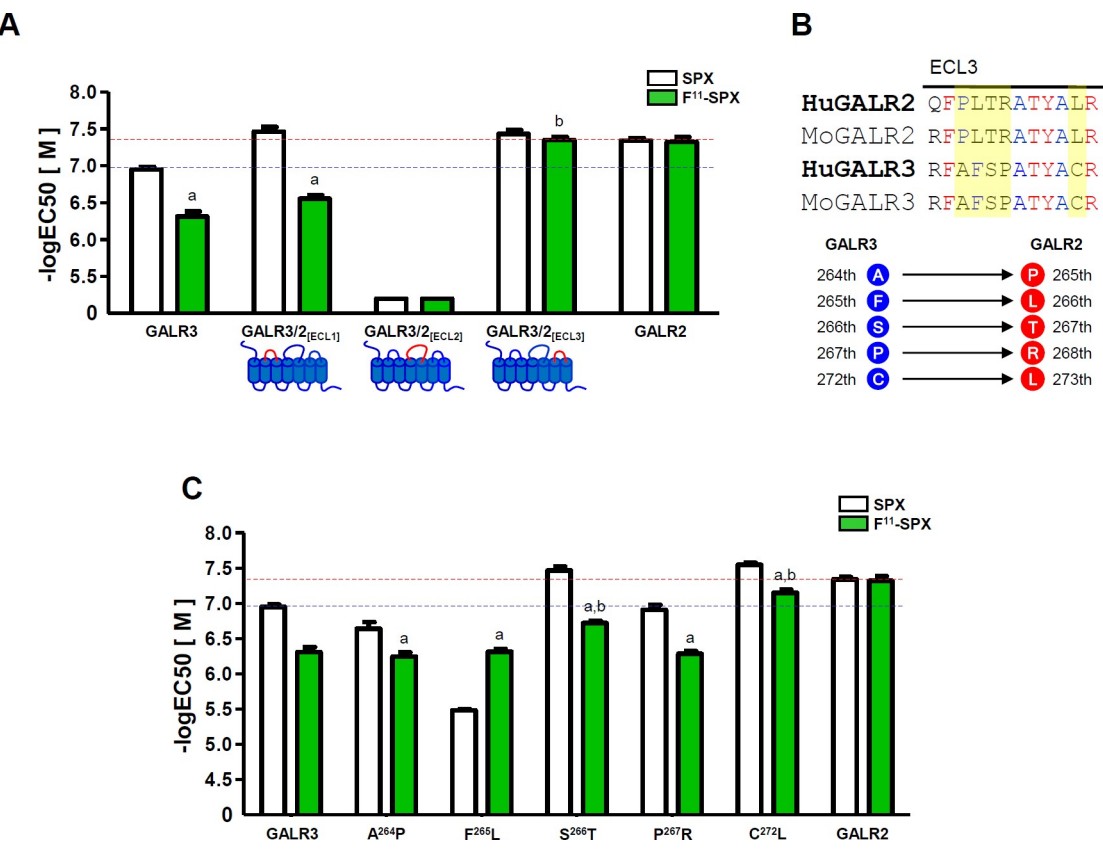

**Fig 6. Identification of core amino acid residues of the ECL domain that are important for interaction with Phe[11] of the ligand.** (A) Responses of the ECL-chimeric receptors to [F[11]]-SPX. (B) Amino acid sequence comparison of the ECL3 domains between human (Hu) and mouse (Mo) GALR2 and GALR3. (C) Responses of single-substitution ECL3 mutants to [F[11]]-SPX. The horizontal red and blue dashed lines represent the responses of GALR2 and GALR3, respectively, to SPX. The $EC_{50}$ values are presented as the mean ± SE (a: $P < 0.05$ vs. SPX; b: $P < 0.05$ vs. WT GALR3).

define the amino acid residues of the GAL peptide that are responsible for receptor binding and activation, alanine (Ala) mutagenesis screening of GAL (2–11) was examined, revealing that Trp[2], Asn[5], Gly[8], and Typ[9] of GAL are crucial for high-affinity binding to GALR2 [31]. A subsequent study suggested that the Trp[2] residue of the GAL (2–11) interacts with conserved histidine residues of the TM6 domain of GALR2, and the Tyr[9] residue of GAL (2–11) exhibited comparable affinity for the ECL3 domain of the same receptor [32]. Because both Trp[2] and Tyr[9] are conserved between GAL and SPX [14], these results showed that the ligand-receptor interaction occurs at least partly at evolutionarily conserved domains. In addition, studies using site-directed mutagenesis and/or molecular docking showed that the molecular interactions of GALRs with GAL involve residues that are conserved among the three GALR subtypes [33, 34]. Thus, previous studies identified the residues responsible for overall receptor activation, but they did not show which residues are responsible for the ligand specificity of the GALR subtypes.

Paralogous genes are produced by gene or chromosome duplications followed by a diversification process involving nucleotide mutations, which eventually leads to the emergence of paralogous proteins that are functionally different yet related to each other. Those events occur for both ligand and receptor gene families, leading to the expansion and co-evolution of ligand-receptor gene families under evolutionary pressure [8]. Thus, changes in the amino

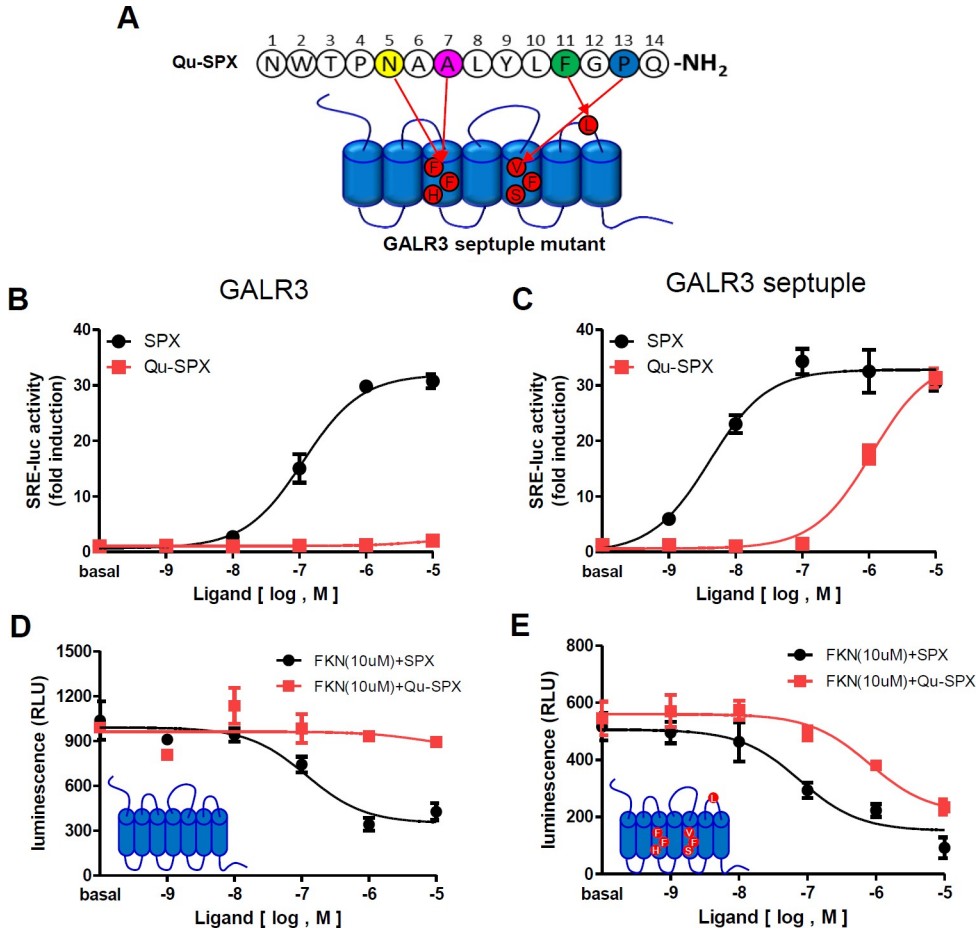

**Fig 7. Effect of the GALR2-specific agonist on the GALR3 septuple-mutant.** (A) Diagram of the GALR2-specific agonist (Qu-SPX) and the construction of the GALR3 septuple-mutant receptor. (B–C) SRE-luc activity in response to increasing concentrations of SPX or Qu-SPX in cells expressing (B) WT GALR3 or (C) the GALR3 septuple-mutant receptor. (D–E) Inhibition of forskolin-induced cAMP production in response to increasing concentrations of SPX or Qu-SPX in cells expressing (D) WT GALR3 or (E) the GALR3 septuple-mutant receptor.

acid sequence of a ligand peptide accompany corresponding changes in the amino acid sequence of a receptor and vice versa [35]. In GAL-SPX evolution, mutations at positions 5, 7, 11, and 13 in the SPX peptide are critical for that peptide to acquire selective affinity for GALR3 [14, 26]. Indeed, the introduction of GAL-originated residues such as Asn[5], Ala[7], Phe[11], and Pro[13] into an SPX-based agonist (Qu-SPX) interferes with the ability of the agonist to interact with GALR3, thus making the agonist specific for GALR2 [15, 26]. However, changes in the GALR3 amino acid sequence that result in high-affinity binding to SPX while decreasing the affinity for GAL have not been addressed.

We hypothesized that there must be key amino acid residues within GALR2 that promote interaction with the Asn[5], Ala[7], Phe[11], and Pro[13] residues of Qu-SPX, whereas other residues in GALR3 attenuate interaction with the same residues of Qu-SPX. We first demonstrated that the Asn[5] and Ala[7] residues of Qu-SPX might favor a conformation formed by the Phe[103], Phe[106], and His[110] residues of the GALR2 TM3 domain but discriminate against the conformation formed by the Leu[100], Tyr[103], and Tyr[107] residues of the GALR3 TM3 domain. Similarly, the Pro[13] residue of Qu-SPX prefers a conformation evoked by the Val[193], Phe[194], and Ser[195] residues in the GALR2 TM5 domain but not the conformation evoked by the corresponding

residues in GALR3. Finally, we found that Phe[11] of Qu-SPX might interact with the Leu[273] residue in the ECL3 domain GALR2 but not with the Cys[272] residue at the corresponding position in GALR3.

Our results showed that WT GALR3 containing the Leu[100], Tyr[103], and Tyr[107] residues responded poorly to the [N[5]]-/[A[7]]-SPX. By contrast, a GALR3 variant in which those residues were substituted with the GALR2-specific residues Phe[103], Phe[106], and His[110], respectively, responded better to [N[5]]-/[A[7]]-SPX. In an earlier report, the Asn[5] residue of GAL was shown to be important for GALR2 binding [31]. Furthermore, the involvement of the above-mentioned amino acid positions in the interactions of GALR2 and GALR3 with ligands has been documented. For instance, an Ala substitution for Phe[106] in GALR2 did not affect GAL binding [32], whereas a mutation (Tyr[103]Ala) in GALR3 abolished binding to GAL [34]. Our results showed that the Tyr[103]Phe mutation alone in GALR3 did not affect the response of that receptor to [N[5]]-SPX or [A[7]]-SPX. A triple (Leu[100]Phe, Tyr[103]Phe, and Tyr[107]His) mutation in the TM3 domain of GALR3 increased ligand binding, however, suggesting that those three amino acids within the TM3 domain of GALR2 interact cooperatively with the Asn[5] and Ala[7] residues of the GALR2-specific agonist.

The Pro[13] residue of GAL and the GAL-like peptide (GALP) is highly conserved across vertebrate species [14]. That residue is thought to be important for a conformation that influences the flexibility of the peptide structure [36]. However, the amino acid at the corresponding position in the SPX peptide is highly variable and can be any one of Ala, Thr, Arg, and Lys [14]. Thus, it seems likely that the Pro[13] residue in Qu-SPX is critical for the binding and activation of GALR2 but hampers interaction with GALR3. Our results suggest that the Pro[13] residue in Qu-SPX might allow high-affinity binding to GALR2 through interaction with the Val[193], Phe[194], and Ser[195] residues in the TM5 domain of the receptor, as the substitution of those residues at the corresponding positions in GALR3 greatly improved the response of that receptor to [P[13]]-SPX.

The residues at position 11 of the GAL/SPX family peptides differ in their biochemical properties: GAL has a hydrophobic Leu, whereas SPX has a basic Lys [14]. Substitution of either Leu or Phe for the Lys[11] residue of SPX significantly decreases the potency of that peptide toward GALR3 but does not affect the potency toward GALR2 [26], which suggests that the presence of a hydrophobic residue at position 11 might contribute to GALR2 selectivity. In addition, the size of the hydrophobic side chain of the residue at position 11 seems to be important because Ala substitution at that position in GAL lowered the binding affinity for GALR2 [37]. Our results showed that [F[11]]-SPX exhibited increased potency toward a GALR3 mutant receptor in which Cys[272] of the ECL3 domain was changed to Leu, suggesting that hydrophobic interaction between Phe[11] of the ligand and Leu[273] of GALR2 might further consolidate the ligand-receptor interaction.

One interesting observation of our study was that Qu-SPX was able to activate a GALR3 mutant receptor in which seven amino acid residues were replaced with the corresponding residues from GALR2. GALR2 and GALR3 exhibit a high degree (64%) of amino acid sequence identity, indicating that they retain a similar topology to build a ligand binding pocket, although their actual ligand binding sites are different. Therefore, it can be postulated that divergence between GAL and SPX at the peptide level has occurred through changes at amino acid positions 5 (Gln↔Asn), 7 (Met↔Ala), 11 (Lys↔Leu), and 13 (Ala↔Pro), while divergence between GALR2 and GALR3 has mainly occurred through Phe[103]↔Leu[100], Phe[106]↔Tyr[103], and His[110]↔Tyr[107] mutations in the TM3 domain; Val[193]↔Ala[190], Phe[194]↔Ala[191], and Ser[195]↔Gly[192] mutations in the TM5 domain; and Leu[273]↔Cys[272] mutation in the ECL3 domain. That divergence process has caused GALR3 to favor SPX and lose its

affinity for GAL, which suggests a strategy for the future development of SPX-based GALR2-specific agonists.

## Conclusion

Our results explain how an SPX-based GALR2 agonist (SG2A) achieves selective interaction with GALR2 while inhibiting interaction with GALR3. Domain swapping and site-directed mutagenesis between GALR2 and enabled us to identify residues in GALR2 that specifically interact with SG2A and, conversely, residues in GALR3 that inhibit interaction with SG2A. When the molecular structure of GALR2/GALR3 becomes available, our findings can be used to support *in silico* virtual screening of small molecules for the development of GALR subtype-specific agonists and antagonists.

## Supporting information

**S1 Text. Supplementary materials and methods** [15, 38]**.**
(DOCX)

**S1 Fig. Membrane expression of the GALR2/3 and GALR3/2 chimeric receptors.** SmBit-tagged chimeric receptors were expressed and treated with LgBiT. Bioluminescence was measured in cells expressing WT GALR2, WT GALR3, chimeric GALR2/3 (A), or chimeric GALR3/2 (B) receptors. Data are presented as the mean ± SE.
(TIF)

**S2 Fig. Molecular dynamics-resolved structures of SPX and Qu-SPX peptides.** Solution structures of SPX (A) and Qu-SPX (B) are predicted using a molecular dynamics (MD) simulation method. MD trajectory analysis was used for the clustering of peptides. Structures with more than 10 frequencies out of 1000 snapshots are displayed.
(TIF)

**S1 Table. Differential responses of GALR2/3 and GALR3/2 chimeric receptors to ligands.**
(DOCX)

**S2 Table. Responses of GALR3-like chimeric receptors to single amino acid-substituted mutant peptides.**
(DOCX)

**S3 Table. Reponses of GALR3 mutant receptors to $N^5$-mutant and $A^7$-mutant peptides.**
(DOCX)

**S4 Table. Responses of GALR3 mutant receptors to $F^{11}$-mutant and $P^{13}$-mutant peptides.**
(DOCX)

**S5 Table. Responses of GALR3 mutant receptors to $F^{11}$-mutant peptide.**
(DOCX)

**S6 Table. Responses of the GALR3 septuple-mutant receptor to Qu-SPX.**
(DOCX)

## Author Contributions

**Conceptualization:** Yoo-Na Lee, Jae Young Seong.

**Data curation:** Yoo-Na Lee, Arfaxad Reyes-Alcaraz, Seongsik Yun, Cheol Soon Lee, Jong-Ik Hwang, Jae Young Seong.

**Formal analysis:** Yoo-Na Lee, Arfaxad Reyes-Alcaraz, Seongsik Yun, Cheol Soon Lee, Jae Young Seong.

**Investigation:** Yoo-Na Lee, Arfaxad Reyes-Alcaraz, Seongsik Yun, Cheol Soon Lee.

**Methodology:** Yoo-Na Lee, Arfaxad Reyes-Alcaraz, Seongsik Yun, Cheol Soon Lee, Jong-Ik Hwang, Jae Young Seong.

**Project administration:** Yoo-Na Lee, Arfaxad Reyes-Alcaraz, Seongsik Yun, Jong-Ik Hwang, Jae Young Seong.

**Supervision:** Jae Young Seong.

**Validation:** Yoo-Na Lee, Arfaxad Reyes-Alcaraz, Jae Young Seong.

**Visualization:** Yoo-Na Lee, Jae Young Seong.

**Writing – original draft:** Yoo-Na Lee, Jae Young Seong.

**Writing – review & editing:** Yoo-Na Lee, Arfaxad Reyes-Alcaraz, Seongsik Yun, Cheol Soon Lee, Jong-Ik Hwang, Jae Young Seong.

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
