## [Decision Letter · Decision Letter 0]

31 Jan 2020

PONE-D-19-33974

Exploring the molecular structures that confer ligand selectivity for galanin type II and III receptors

PLOS ONE

Dear Dr Seong,

Thank you for submitting your manuscript to PLOS ONE. After careful consideration, we feel that it has merit but does not fully meet PLOS ONE’s publication criteria as it currently stands. Therefore, we invite you to submit a revised version of the manuscript that addresses the points raised during the review process.

Based on the comments of the two reviewers and on my own reading, the manuscript is deemed scientifically sound but you should pay close attention to all the issues raised by the reviewers. In particular, it is noted that the manuscript reports on a solid molecular biology study but the variability of the conformational profile is not considered in discussing the results. This point is crucial and should be carefully discussed. Also, the possibility to conduct complementary analysis, like NMR, in order to check that the profile does not change should be taken into account.

We would appreciate receiving your revised manuscript by March 2, 2020. To enhance the reproducibility of your results, we recommend that if applicable you deposit your laboratory protocols in protocols.io, where a protocol can be assigned its own identifier (DOI) such that it can be cited independently in the future. For instructions see: http://journals.plos.org/plosone/s/submission-guidelines#loc-laboratory-protocols

We look forward to receiving your revised manuscript.

Kind regards,

Pere Garriga

Academic Editor

PLOS ONE

Journal Requirements:

Reviewers' comments:

Reviewer's Responses to Questions

**Comments to the Author**

1. Is the manuscript technically sound, and do the data support the conclusions?

Reviewer #1: Yes

Reviewer #2: Yes

2. Has the statistical analysis been performed appropriately and rigorously? 

Reviewer #1: Yes

Reviewer #2: Yes

3. Have the authors made all data underlying the findings in their manuscript fully available?

Reviewer #1: Yes

Reviewer #2: Yes

4. Is the manuscript presented in an intelligible fashion and written in standard English?

Reviewer #1: No

Reviewer #2: Yes

5. Review Comments to the Author

Reviewer #1: This is an interesting manuscript that tries to elucidate the molecular specificities of galanin type 2 and 3 receptors for their ligands. The authors should be commended for the large body of work done and using different techniques.

I have some suggestions that can improve the manuscript.

1. English needs significant improvement. For example, in the abstract they state "we identified residues in GALR2 that "attract" SG2A and residues in GALR3 that "avoid" SGA2. How can amino acids "attract" or "avoid" ligands? Suggest using scientific terms such as interact, inhibit etc.,

2. Should show sequence of GAL peptide in Figure 1.

3. How similar is the Qu-SPX peptide compared to GAL? Is SPX different from GAL peptide in only the four amino acids replaced or would more complete substitution with amino acids from GAL peptide in Qu-SPX rescue GALR3 activity for the Qu-SPX peptide?

This might explain why the GALR3 septuple mutant shows very subtle activity towards Qu-SPX.

Reviewer #2: The authors in the present work, report the results of a study aimed at identifying residues of the GAL2R and GalR3 galanin receptors that permit ligand selectivity of a spexin analog. For this purpose, the authors carry out GAL2R and GAL3R receptor swapping and site directed mutagenesis studies.

The main flaw of the manuscript regards the interpretation of the results. In the manuscript, the discussion is carried out considering that the observed differences in the binding affinity of the diverse analogs can be interpreted in terms of ligand-receptor interactions. However, peptides are flexible and their conformational profile is affected by the sequence. We may consider that one amino acid substitution may not change much the conformational profile of the analogs, but four as in Qu-SPX is risky. Accordingly, the discussion should be carried out considering a loss of an interaction and/or to a different ligand conformational behavior. These considerations are lacking in the present manuscript.

Minor details:

The sentence: “The strong response to WT SPX was retained by all of the GALR2/3 series a to c chimeric receptors, whereas it was drastically weakened in the GALR2/3 series d and e chimeric receptors” requires revision.

I suggest referring to SPX in the same way all through the manuscript. For example, WT SPX

6. PLOS authors have the option to publish the peer review history of their article (what does this mean?). If published, this will include your full peer review and any attached files.

Reviewer #1: No

Reviewer #2: No

---

## [Author Response · Author response to Decision Letter 0]

29 Feb 2020

Dear Editor

We thank you for allowing us to revise our manuscript entitled “Exploring the molecular structures that confer ligand selectivity for galanin type Ⅱ and Ⅲ receptors” [PONE-D_19_33974]. We carefully revised the manuscript to incorporate comments and concerns raised by the editor and reviewers. The major changes are as follows.

1. To address the comment from the editor and reviewer 2, we conducted molecular dynamics (MD) simulation study to predict the solution structures of SPX and Qu-SPX. MD-resolved structures of SPX and Qu-SPX differ each other, although both peptides may not have stable three-dimensional structures in solution. This result raises a possibility that conformations of Qu-SPX different from those of SPX may hamper full recovery of the potency toward the septuple-mutant GALR3. The result is shown in S2 Fig. As this experiment was conducted by Dr. Chul-Soon Lee we (all authors) decided to include him as a co-author of the article. 

2. The GAL sequence is included in Fig 1 as reviewer 1 suggested.

3. The manuscript is revised to meet PLOS ONE's style requirements. 

Response to reviewer #1

We appreciate your valuable and helpful comments on our manuscript. 

Point 1: English needs significant improvement. For example, in the abstract they state "we identified residues in GALR2 that "attract" SG2A and residues in GALR3 that "avoid" SGA2. How can amino acids "attract" or "avoid" ligands? Suggest using scientific terms such as interact, inhibit etc.

→We used scientific terms in the revised manuscript. Please see page 2 line26, and line 31. Please see page 4 line 86 line. Please see page 17 line 465, and 466. 

Point 2 : Should show sequence of GAL peptide in Figure 1.

→We added the GAL peptide sequence in the revised Figure 1 (please see Page 5 line 102 and line 104). 

Point 3 : How similar is the Qu-SPX peptide compared to GAL? Is SPX different from GAL peptide in only the four amino acids replaced or would more complete substitution with amino acids from GAL peptide in Qu-SPX rescue GALR3 activity for the Qu-SPX peptide? This might explain why the GALR3 septuple mutant shows very subtle activity towards Qu-SPX.

→ We agree on reviewer’s opinion that additional substitutions with amino acids from GAL peptide in Qu-SPX rescue GALR3 activity more. In our previous report (Reyes-Alcaraz et al., 2016, Ref# 26), we explained in detail that four amino acids replacement (Asn5, Ala7, Leu11/Phe11, and Pro13 substitution) in SPX completely abolished potency to GALR3 while mutations in other residues of SPX did not influence GALR2/GALR3 selectivity. Therefore, we focused on these 4 residue substitutions. We include these points in the revised manuscript. (Please see page 4 line 80). 

In addition, we conducted molecular dynamics (MD) simulation analysis to predict the solution structures of SPX and Qu-SPX. The result shows that structures of SPX and Qu-SPX differ each other, although both peptides may not have stable three-dimensional structures in solution. This result raises a possibility that conformations of Qu-SPX different from those of SPX may hamper full recovery of the potency toward the septuple-mutant GALR3. This notion is included in the revised text. Please see S1 text for Molecular Dynamics and S2 Fig.

Response to reviewer #2

We appreciate your valuable and helpful comments on our manuscript. 

Point 1: The main flaw of the manuscript regards the interpretation of the results. In the manuscript, the discussion is carried out considering that the observed differences in the binding affinity of the diverse analogs can be interpreted in terms of ligand-receptor interactions. However, peptides are flexible and their conformational profile is affected by the sequence. We may consider that one amino acid substitution may not change much the conformational profile of the analogs, but four as in Qu-SPX is risky. Accordingly, the discussion should be carried out considering a loss of an interaction and/or to a different ligand conformational behavior. These considerations are lacking in the present manuscript

→ To address this point, we tried to predict the solution structures of SPX and Qu-SPX using a molecular dynamics (MD) simulation method. MD trajectory analysis was used for the clustering of peptides. Indeed, MD-resolved structures of SPX and Qu-SPX differ each other, although both peptides may not have stable three-dimensional structures in solution. This result raises a possibility that conformations of Qu-SPX different from those of SPX may hamper full recovery of the potency toward the septuple-mutant GALR3. This notion is included in the revised text. Please see S1 text for Molecular Dynamics and S2 Fig.

Minor details: “The strong response to WT SPX was retained by all of the GALR2/3 series a to c chimeric receptors, whereas it was drastically weakened in the GALR2/3 series d and e chimeric receptors” requires revision. I suggest referring to SPX in the same way all through the manuscript. For example, WT SPX

→ WT SPX is changed to SPX throughout the revised manuscript.

---

## [Editor Report · Decision Letter 1]

11 Mar 2020

Exploring the molecular structures that confer ligand selectivity for galanin type II and III receptors

PONE-D-19-33974R1

Dear Dr. Seong

We are pleased to inform you that your revised manuscript, after taking into account all the reviewers comments, has been judged scientifically suitable for publication and will be formally accepted for publication once it complies with all outstanding technical requirements.

With kind regards,

Pere Garriga

Academic Editor

PLOS ONE
---

## [Editor Report · Acceptance letter]

16 Mar 2020

PONE-D-19-33974R1 

Exploring the molecular structures that confer ligand selectivity for galanin type II and III receptors 

Dear Dr. Seong:

I am pleased to inform you that your manuscript has been deemed suitable for publication in PLOS ONE. Congratulations! Your manuscript is now with our production department. 

With kind regards,

on behalf of

Dr. Pere Garriga 

Academic Editor

PLOS ONE